# How many people need to classify the same image? A method for optimizing volunteer contributions in binary geographical classifications

Carl Salk[1,2]*, Elena Moltchanova[3], Linda See[4], Tobias Sturn[4], Ian McCallum[4], Steffen Fritz[5]

1 Southern Swedish Forest Research Centre, Swedish University of Agricultural Sciences, Alnarp, Sweden, 2 Faculty of International Studies, Utsunomiya University, Utsunomiya, Japan, 3 School of Mathematics and Statistics, University of Canterbury, Christchurch, New Zealand, 4 Novel Data Ecosystems for Sustainability Group, Advancing Systems Analysis Program, International Institute for Applied Systems Analysis, Laxenburg, Austria, 5 Strategic Initiatives Program, International Institute for Applied Systems Analysis, Laxenburg, Austria

* carl.salk@slu.se

**Data Availability Statement:** The dataset analyzed in this study archived in the Zenodo repository and is available at doi.org/10.5281/zenodo.5986721.

## Abstract

Involving members of the public in image classification tasks that can be tricky to automate is increasingly recognized as a way to complete large amounts of these tasks and promote citizen involvement in science. While this labor is usually provided for free, it is still limited, making it important for researchers to use volunteer contributions as efficiently as possible. Using volunteer labor efficiently becomes complicated when individual tasks are assigned to multiple volunteers to increase confidence that the correct classification has been reached. In this paper, we develop a system to decide when enough information has been accumulated to confidently declare an image to be classified and remove it from circulation. We use a Bayesian approach to estimate the posterior distribution of the mean rating in a binary image classification task. Tasks are removed from circulation when user-defined certainty thresholds are reached. We demonstrate this process using a set of over 4.5 million unique classifications by 2783 volunteers of over 190,000 images assessed for the presence/absence of cropland. If the system outlined here had been implemented in the original data collection campaign, it would have eliminated the need for 59.4% of volunteer ratings. Had this effort been applied to new tasks, it would have allowed an estimated 2.46 times as many images to have been classified with the same amount of labor, demonstrating the power of this method to make more efficient use of limited volunteer contributions. To simplify implementation of this method by other investigators, we provide cutoff value combinations for one set of confidence levels.

**Funding:** The funding from IIASA provided support in the form of salaries for one authors [CS], but neither IIASA nor the European Space Agency had any additional role in the study design, data collection and analysis, decision to publish, or preparation of the manuscript.

**Competing interests:** The authors have declared that no competing interests exist.

# Introduction

Since J. Howe coined the term 'crowdsourcing' [1], there has been a notable rise in the use of outsourcing a range of tasks to the crowd by organizations that do not have the labor to carry out these tasks themselves. In many commercial contexts, the completion of these tasks is tied to micropayments, e.g., through the use of Amazon's Mechanical Turk [2]. Such contributions can be valuable; for instance an online crowd can make better predictions of stock performance than the stock market index [3]. Crowdsourcing can also contribute to many aspects of science. This is known as 'citizen science', which is generally defined as the involvement of citizens in scientific research and knowledge production [4, 5]. When it deals with spatially explicit information, crowdsourcing is sometimes known as volunteered geographic information (VGI [6]). Crowdsourcing and VGI are at the lowest level in Haklay's [7] typology of participation in citizen science, where data collection is one of the most common and basic tasks. Data collection ranges from field-based activities, e.g., bird watching [8], to online activities such as transcription of documents [9], mapping for humanitarian causes through VGI initiatives such as OpenStreetMap [10] to image classification. Examples of the latter include the highly successful Zooniverse in which participants classify galaxies [11], Geo-Wiki, where satellite imagery is interpreted for land cover and land use [12], and various citizen science projects that involve tasks like identifying species from camera trap photos [13, 14], among others.

The quality of data from crowdsourcing and citizen science has recently become an active area of research [15, 16]. Many different methods for quality assurance have been developed including, among others, automated checking [17, 18], comparison with an expert or gold standard data sets [19], comparison with an alternative source of data as a proxy [17, 18], crowdsourced and expert peer review [19] and assigning multiple volunteers to perform the same task [20, 21]. When using multiple volunteers, assigning higher weighting to volunteers who performed better individually resulted in better aggregate performance [3]. When the multiple-volunteer approach is used, key questions include how to combine the observations and how much certainty they provide about the actual phenomenon being investigated. Haklay et al. [22] first researched this question using data from OpenStreetMap, framing it in terms of Linus' Law from open-source coding, i.e., the more contributors to the code, the higher the quality. The authors applied this to the positional accuracy of roads in OpenStreetMap in the UK and found that the first five contributors provided the largest contributions to improving the accuracy while there was little to be further gained beyond 13 contributors. A similar question was asked by Hsing et al. [14] about classifying pictures of mammals from camera traps, motivated by the need to use scarce volunteer effort more efficiently to process the deluge of photographs generated. Using data from the MammalWeb project in the UK, they found that 99% of classifications were correct when at least nine volunteers provided the same answer. However, they also found large variations in this pattern across species. Siddharthan et al. [23] developed a Bayesian consensus model to minimize crowd size for the task of assigning a photo to one of 22 possible bumblebee species; they demonstrated this to be more efficient than either majority voting or the use of fixed threshold. These approaches have clear implications for image classification activities that either have an associated cost, limited expert peer review or where the supply of photographs vastly outstrips the crowdsourcing demand.

In this paper we develop a method to decide when to remove tasks from circulation to volunteer raters and apply it to a case study of image classifications from the Cropland Capture game in which participants classified satellite imagery and photographs as cropland or noncropland, including a 'maybe' option if unsure. Previous research has shown the limitations of using majority voting for classifying images in this type of task [24]. The approach presented

here builds on this work to determine when a binary classification task has reached an acceptable level of certainty and further classifications are unlikely to change the outcome. This allows tasks reaching this threshold to be removed from the pool assigned to volunteers, thereby freeing up labor that can be applied to additional tasks, optimizing the use of the crowd.

## Theoretical approach

Determining whether enough ratings have been performed to remove an individual classification task (i.e. an image) from circulation (either because a strong agreement has been reached, or because the crowd is deadlocked and agreement is unlikely to be reached) requires a decision based on at least two factors which we here refer to as (1) the maximum acceptable disagreement rate from the classification of the majority of the crowd, and (2) the degree of certainty that the crowd is within that disagreement level (see graphical explanation in Fig 1A). Both of these factors need to be chosen by organizers of the activity based on the needs of their project.

The disagreement rate is simply the percentage of user classifications that disagree with the majority classification. Consider a hypothetical land cover classification task where four volunteers say they see cropland and a fifth does not detect cropland. In this case, 20% of the total

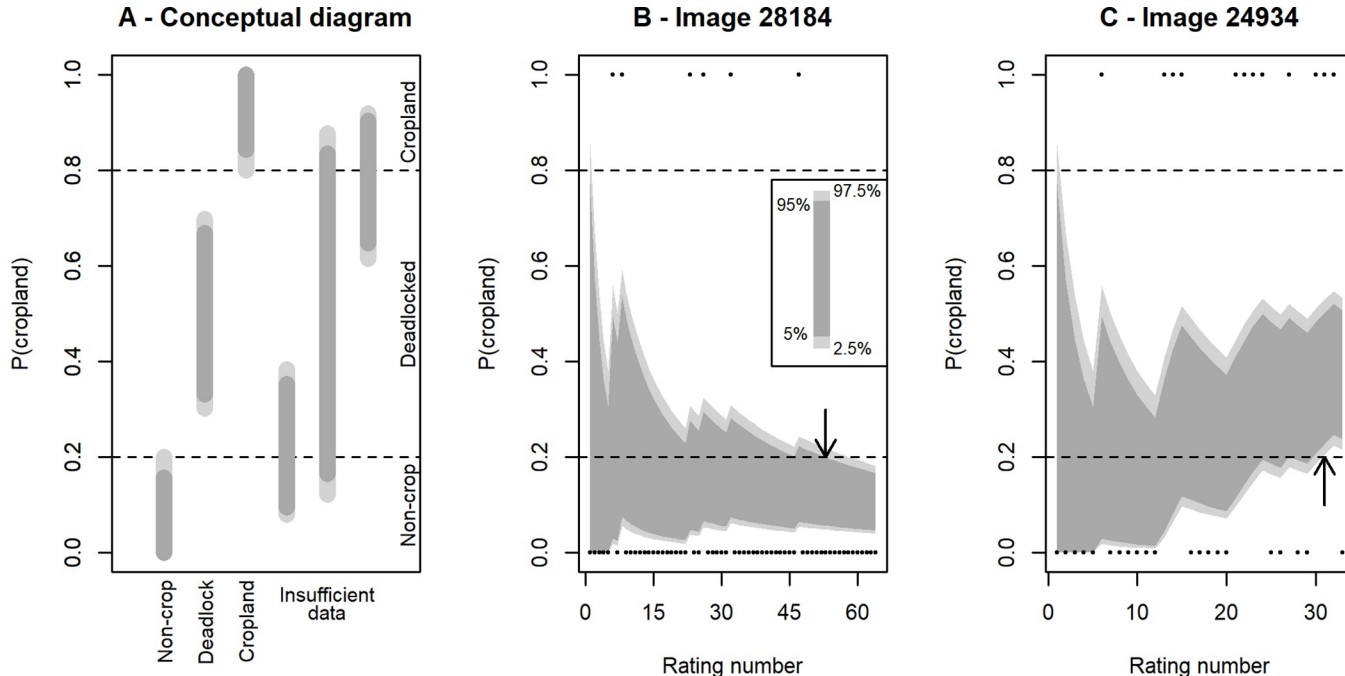

**Fig 1. A graphical illustration of our proposed method to efficiently decide that an image is sufficiently well classified.** When one of the conditions illustrated here are met, the task can be pulled from circulation as a volunteer classification task. Panel (A) demonstrates the decision-making mechanism. The inset in panel B shows how the different shading intensities indicate different quantile levels; the two shades of gray have identical meanings in all three panels. If the 95th percentile of the posterior distribution is below the selected disagreement rate (20% in our case—horizontal dashed lines), then the image is concluded to contain no cropland. Similarly, if the 5th percentile is above 80% cropland, then we conclude that it does contain cropland. Note that in both of these cases, it is fine if the wider (light gray) interval still overlaps the target disagreement rate. If the central 95% (i.e. 2.5% to 97.5%) of the posterior distribution is entirely between 20% and 80%, the image is pulled from circulation because the crowd is unlikely ever to reach a definitive conclusion. The right three bars show situations where more data is sought. Panels (B) and (C) depict two applications of this method to real data. The dots along the top and bottom of the panels show actual ratings data. The dark and light gray shaded areas (representing the same ranges as in panel A) show how the estimated probability of cropland vs. non-cropland classifications changes with additional classifications. The arrows show the point at which the decision is made to withdraw an image from circulation. Note that in panel B this decision is made when the 95th percentile (i.e. dark gray region) falls below the target, but that in panel C, the conclusion is only reached when the light gray regions are entirely between the targets.

ratings disagree with the majority. For many applications, this could be seen as a reasonable level of agreement, and is the level that we use in the practical demonstrations presented later in this study. However, these hypothetical ratings are not particularly strong evidence about the correct classification of the image—with only five ratings, this observed disagreement rate could easily be the result of uninformed random guessing. What is important is not simply the underlying rate of disagreeing classifications (which can be estimated as the proportion of observed ratings disagreeing with the majority rating), but also the confidence that the estimated disagreement rate is within the desired range. This leads to the second factor that must be chosen, the desired degree of certainty that the disagreement rate is below the chosen limit. In the analyses below, we choose a 95% certainty level. Taken together, these two choices mean that we decide the collective ratings are sufficient to declare an image classified when we are 95% certain that the mean disagreement rate is below 20% (Fig 1A, 1B). Similarly, an image is declared unlikely to ever be classified when, given the evidence available, we are 95% certain that the mean disagreement rate is above 20% (e.g. Fig 1A, 1C). Estimating the probability distribution of the disagreement rate is a Bayesian calculation. In Bayesian analysis, the observations (data) are combined with the prior belief about the process, expressed via a probability distribution, to estimate a posterior distribution for the parameter of interest. Here, the outcome of the analysis is the posterior distribution of the disagreement rate. The prior distribution reflects our state of knowledge before the experiment. In the absence of such knowledge, it can be weak, (i.e. minimally-informative), meaning that the posterior distribution is virtually uninfluenced by it once even a small amount of data is accumulated.

Our proposed system functions as follows. In general, consider an image which has been rated $n_c$ times as 'cropland' and $n_{nc}$ times as 'non-cropland'. Assuming, *a priori* a beta distribution $Beta(\alpha_0, \beta_0)$ for the probability of the image containing cropland, and applying Bayes' theorem, we obtain the posterior distribution $Beta(\alpha = \alpha_0 + n_c, \beta = \beta_0 + n_{nc})$. The average disagreement rate can then be evaluated as $min(\alpha/(\alpha + \beta), \beta/(\alpha + \beta))$, i.e., the percentage of participants disagreeing with the majority vote of either cropland or non-cropland. Let $Q_y$ denote the $\gamma^{th}$ quantile of the posterior beta distribution for some $0 \leq \gamma \leq 1$ (this is what is called the certainty rate in the previous paragraphs). We then define the following two outcomes our of interest: (i) A sufficient number of votes has been cast when we are at least $\gamma \times 100\%$ certain that the disagreement rate is below some disagreement rate threshold $D$ $(D < 50\%)$. In other words, either $Q_y \leq D$ or $Q_{y-1} \geq 1 - D$, or (ii) A deadlock has been reached, when we are $\gamma \times 100\%$ certain that the disagreement rate is above $D$. In other words, $D < Q_{(1-\gamma)/2} < Q_{1-(1-\gamma)/2} < 1—D$.

In the following demonstration of this approach, we have chosen prior parameters of $\alpha_0 = 0.5$ and $\beta_0 = 0.5$, a disagreement rate of $D = 20\%$ and a degree of certainty of $\gamma = 95\%$. The prior parameters make for a weak prior with minimal influence on the eventual decision to declare an image classified or unclassifiable. This prior treats the two classification outcomes (cropland or non-cropland) as being equally likely, and more likely than an inconclusive classification. Although beyond the scope of this work, different prior parameters are possible for the beta distribution, for instance to account for an expectation that one category is more common than the other or that strong disagreement among volunteers on a particular task is unlikely.

## Materials and methods

### Data collection campaign

The data analyzed in this study came from the main campaign of the Cropland Capture game conducted between November 2013 and May 2014 which has been described in detail

elsewhere [21, 24] and is publicly accessible at doi.org/10.5281/zenodo.5986721. In summary, the game presented volunteer raters with images (mostly from satellites, but some from ground-based photographs) which they had to classify as including some cropland (no matter how small the area), or containing no cropland. Some example images can be seen in our previous publications [e.g. 21, 24]. If unsure about the classification, raters could choose a third 'maybe' option which is not considered in the calculations described in the previous section or in the results section. Independence among raters was ensured by not providing information about previous classifications. Further, the sheer number of images and lack of unique image identifiers accessible to the raters essentially eliminated any possible impacts of raters discussing individual images. A total of 2,783 volunteers contributed a total of 4,461,708 ratings of 191,027 images. The volunteers were recruited mostly via appearances of project leaders in English- and German-language media [25]. Thus, many volunteers came from English- and German-speaking countries, although dozens of other countries around the world were represented. Interestingly, the classification skill of raters had very little relationship to whether the images were from their home area [25], so the international distribution of volunteers has little possibility to impact the results presented here.

## Analytical methods

To illustrate the benefits of a systematic procedure for removing images from circulation, we first computed the 95% central posterior credible intervals (i.e. the range of values corresponding to the 2.5% to 97.5% percentiles of the distribution) for the mean rating of all images used in the Cropland Capture campaign. To illustrate the evolution of these intervals as additional data accumulates, we re-computed the image's intervals after each additional rating. This yielded a series of numbers that are visualized as the shaded gray areas in Figs 1B, 1C and 2. All analyses were performed in R version 3.3.2 [26].

Based on these series, we computed distributions for the number of ratings it takes to reach either a definitive classification (cropland, no cropland) or declare it unlikely that a definitive classification will ever be reached (deadlock) as defined above. These distributions were calculated both for the entire image set and for the subsets that were ultimately classified in each of the three possible ways (no cropland, cropland and deadlocked). To evaluate the probability of false positive errors, we calculated the proportion of images that would have been removed from the active pool (according to the standards described above) that would later receive ratings that take them out of the pre-defined range for removal, and also that would eventually receive a different classification. To quantify potential efficiency gains, we calculated the percentage of images that could have been removed from circulation before the end of the Cropland Capture campaign and estimated how many more images could have been classified had this effort been allocated to additional tasks. Finally, we repeated these efficiency gain calculations for the scenario that all images are pulled from circulation after reaching the 95th percentile of the number of ratings for declaring an undecided classification; we chose this group because undecided conclusions were reached more slowly on average than cropland or non-cropland classifications.

As a further analysis of the chances of mis-classification under our proposed system, we simulated random draws of either non-cropland or cropland with probabilities of choosing cropland ranging from $\theta = .01$ to $\theta = .99$ in intervals of .01. Successive random draws continued until one of the three possible conclusions described above was reached. This process was repeated 1000 times for each value of $\theta$, and we recorded the number of ratings to reach a conclusion and the conclusion reached for each individual simulation. Based on these outcomes, we computed the median number of ratings to reach a conclusion, and the probability of incorrect conclusions at each value of $\theta$. Initial simulations showed that reaching a conclusion

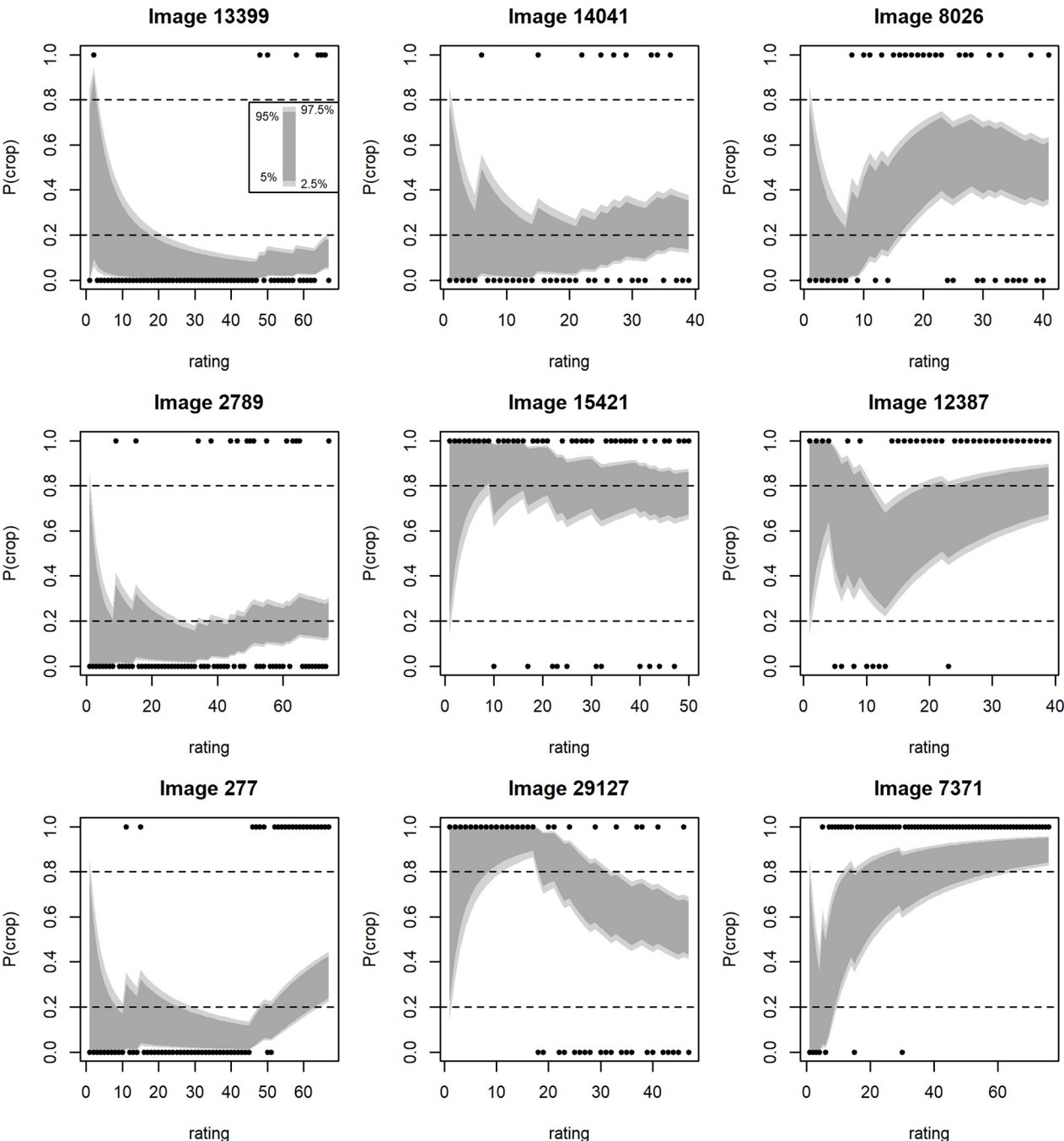

**Fig 2. Some examples of the evolution of the posterior 95% credible interval of the estimated mean rate of classification of an image as cropland by volunteer raters.** Each panel corresponds to a different image; the number of volunteer-contributed ratings differed among images (note the different x-axis scales). The inset in the first panel shows how different gray shades represent different quantile ranges of the estimated mean probability of classifying an image as cropland. The points along the upper and lower margins of panels represent actual ratings (0 = not cropland, 1 = cropland). The horizontal dashed lines represent the level of certainty about classifications (<20% disagreement) considered in this study. Most images show patterns similar to those in the top row of this figure.

when $\theta = .2$ or $\theta = .8$ (the respective cutoffs for reaching non-cropland and cropland conclusions) usually took so long that estimating the median time to conclusion was computationally impractical (in addition to being uninformative for the design of these activities).

## Results

Under our chosen acceptable levels of uncertainty, it took a minimum of nine (unanimous) classifications to reach a conclusion that an image is either cropland or non-cropland, and 10 evenly split (five yes and five no) classifications to conclude that the volunteer crowd is unlikely to ever reach a conclusive decision. The crowd tended to converge on answers fairly quickly. For 93.1% of images, the first rating collected agreed with the eventual majority rating of the crowd for that image. A confident conclusion about an image could be reached with the minimum possible number of ratings for 54.3% of images (Fig 3A). Concluding that an image did not include cropland took on average 10.0 classifications; 95% were concluded with 18 or fewer ratings, and 99% were reached within 33 ratings (Fig 3B). Concluding that an image

### A - All classified images

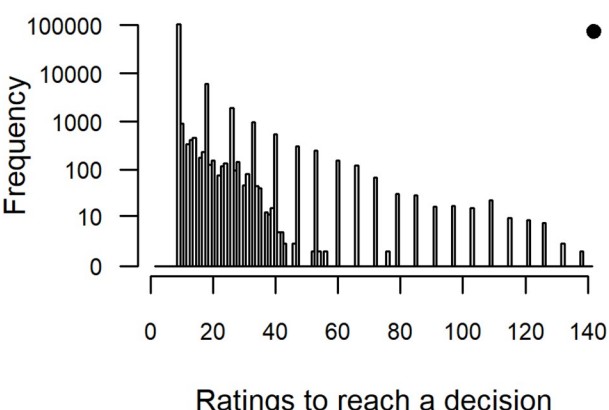

### B - Images classified as non-crop

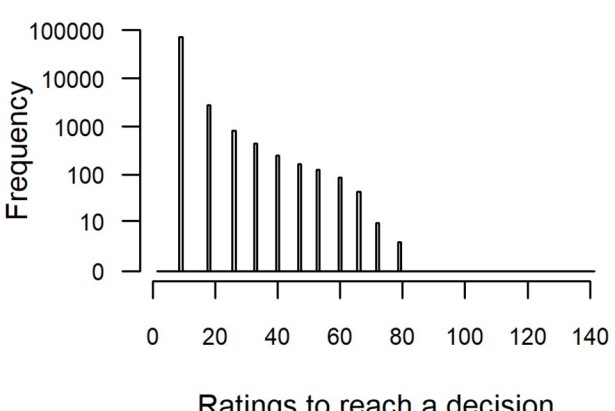

### C - Images classified as cropland

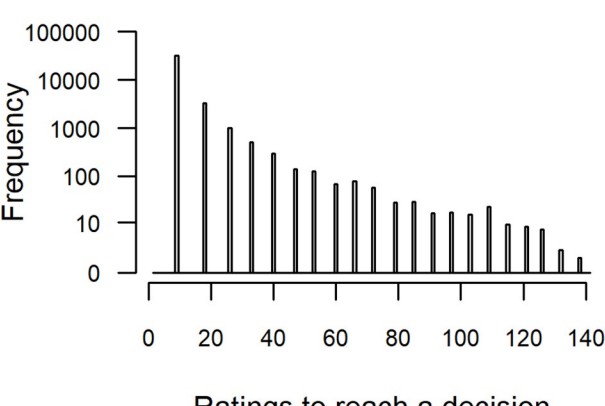

### D - Deadlocked images

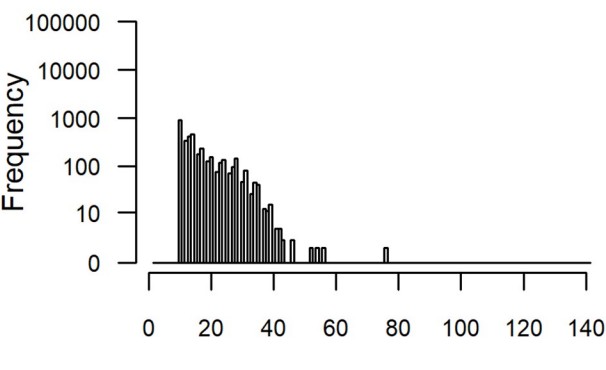

**Fig 3. Distributions of the number of volunteer ratings required to draw a conclusion about the classification of the image.** (A) all images combined, (B) images eventually classified as non-cropland, (C) images classified as cropland, and (D) images where the crowd showed no agreement. The logic of declaring an image concluded is laid out in the Theoretical Approach section. Note the modified log scale on the y-axes. The dot in the upper right corner of panel A shows how many images never reached a conclusive classification. The peaks seen in panels A-C is because the classification ratios are limited by the counts in each category necessarily being integers. So the first peak is the situation where there are no dissenting ratings, the second peak includes one dissent, and so forth.

included cropland was slightly slower. On average, this conclusion required 11.7 classifications to reach, but there was a long tail of slower conclusions; concluding 95% of images required 26 classifications, and 99% completion needed 53 classifications (Fig 3C). Deciding that the volunteer crowd was unlikely to reach a conclusion required on average 16.9 ratings; 95% of inconclusive decisions were reached by 31 ratings, and 99% were reached by 38 ratings (Fig 3D).

Collecting additional data beyond what was required to reach a conclusion about an image's classification sometimes weakened the conclusion when new ratings contradicted the previous trend. The probability of the 95% quantile range growing beyond the required standard after a conclusion was reached was 15.0% for non-cropland images, 18.2% for cropland containing images, and 65.7% for deadlocked images (see examples in the second row of Fig 2). However, examples where an image that would have been pulled from circulation would eventually reach a different classification with more ratings were very rare; this happened to only 0.2% of images initially classified as non-cropland, 0.1% of initially cropland-classified images, and 0.6% of initially deadlocked images (but see examples of these trajectories in the third row of Fig 2). All of the cases where the classification of an image changed with additional data were what we refer to as 'weak changes', meaning there was a switch between cropland or non-cropland and an inconclusive classification, or vice versa. No examples were seen where an image classification changed between cropland and non-cropland with additional classifications.

Applying the decision rules described above would have resulted in 61.2% of the images being removed from circulation before the campaign ended (Fig 4). This approach would have eliminated the need for 59.4% of all ratings performed by volunteers in the campaign analyzed here. Assuming that additional images are similarly difficulty, applying the effort for the unnecessary ratings to new tasks would allow for an estimated $(100\% - 59.4\%)^{-1} = 2.46$ times as many images to be classified with the same amount of volunteer effort. Applying a further restriction of always quitting after 34 ratings (the 95% quantile for number of ratings for concluding undecided images–see first paragraph of Results) leads to only modest additional gains in classification efficiency, increasing the proportion of unneeded ratings by only 1.1 percentage points, up to a value of 60.5%. However, because it takes 36 majority ratings to classify an image with four dissenting ratings (Fig 5), this effectively means that any task that has accumulated four dissenting ratings can immediately be removed from circulation. Had this rule been applied, the proportion of unneeded ratings would have increased slightly further, to 63.6%.

The simulation shows that images were overwhelmingly correctly classified as non-cropland, deadlocked or cropland (Fig 6). Even when the underlying probability of classifying an image as cropland was within 1 percentage point of the cutoffs for non-cropland (.2) and cropland (.8), the probability of incorrect classification was relatively low. For underlying probability of $\theta = .19$, only 12.3% of images were misclassified, all as deadlocked. For $\theta = .81$, a similar value of 13.9% misclassification was observed, again all as deadlocked. For $\theta = .21$ and $\theta = .79$, the most extreme simulated values that should have still resulted in a deadlock, the error rates were 29.8% and 32.7%, respectively (Fig 6). In no case was a misclassification between non-cropland and cropland observed.

## Discussion

This study has developed a system to probabilistically evaluate the true identity of unknown binary classification tasks and decide when further independent volunteer classifications are unlikely to yield new information. We have also demonstrated the benefits of this system, particularly in the realm of volunteered geographical information (VGI), using data from a binary

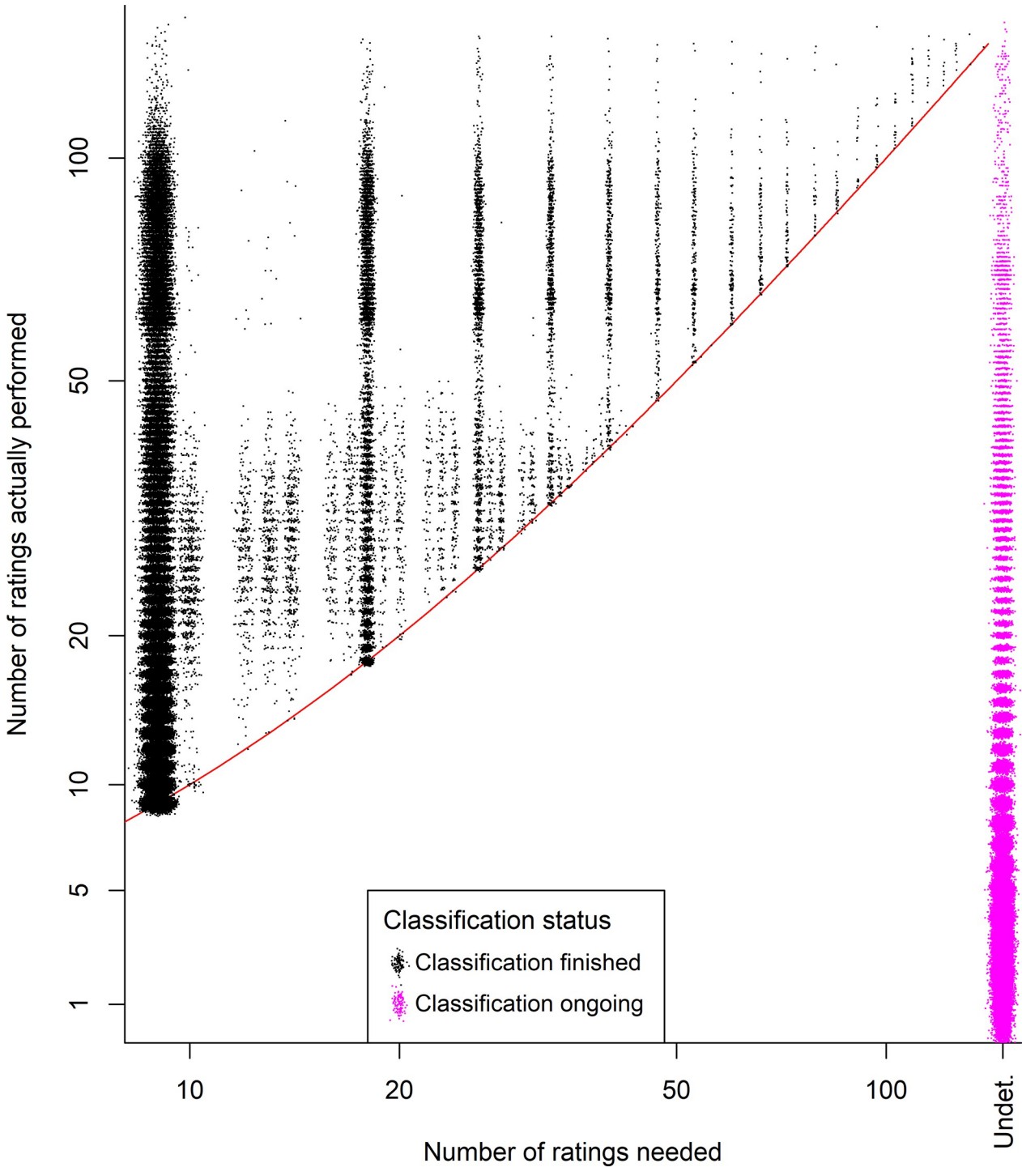

**Fig 4. An illustration of the distributions of how many ratings were needed to classify an image versus how many were actually performed.**
The red line is a 1:1 line showing the number of ratings needed to classify a particular image (it appears slightly curved due to the modified log scale on the y-axis). The distance a black point falls above this line indicates how many unnecessary ratings were performed; some points fall slightly below this line because the points were jittered for clarity. The pink points indicate images that had not yet been classified at the end of the Cropland Capture campaign, thus it is undetermined (Undet.) how many ratings would have been required. Five images with hundreds of ratings each are omitted from this figure.

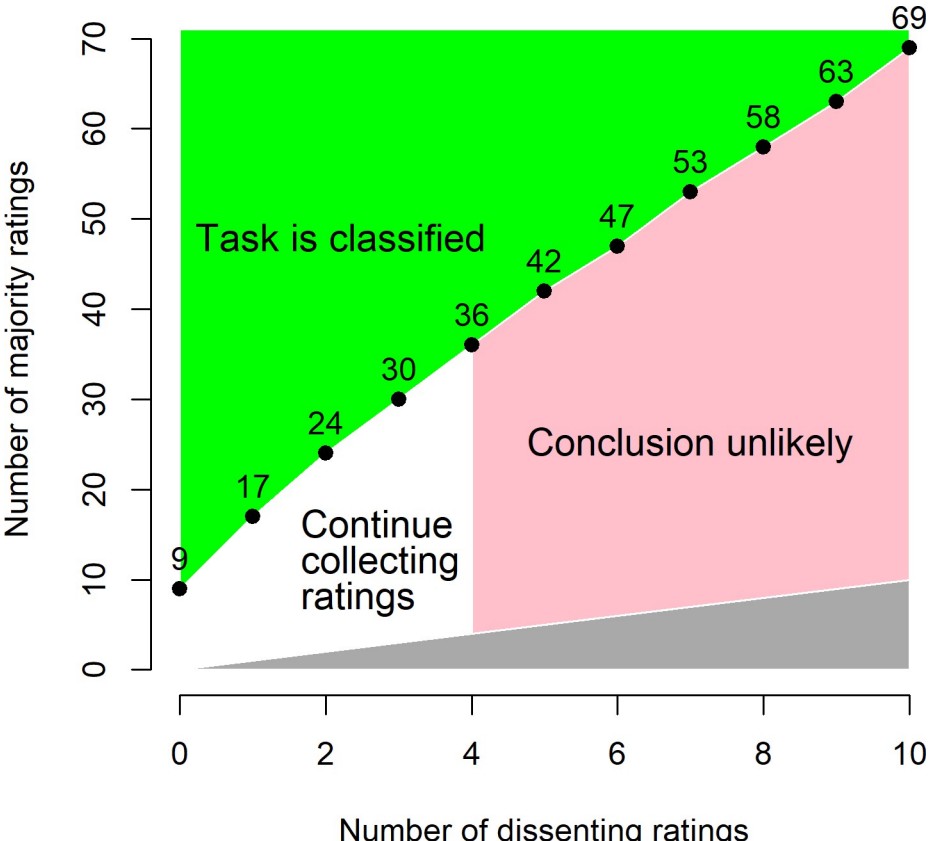

**Fig 5. Conditions for deciding to remove a task from a crowdsourcing campaign according to the system outlined in this paper.** If the number of majority ratings (y-axis) and the number of dissenting ratings (x-axis) fall within the upper (green) region, then the task can be confidently classified according to the majority of ratings. The numbered black dots are shown to facilitate implementation of this system in other settings/software; they indicate the minimum number of majority ratings to fall in the green region as a function of the number of dissenting ratings. If four or more dissenting ratings have been received (pink region to right), then it is unlikely that a definitive conclusion will ever be reached, so collection of further ratings can end and the task might be flagged for assessment by experts. The combination of majority and dissenting ratings can never fall in the lower (gray) area because the number of majority ratings cannot be smaller than the number of dissenting ratings. These calculations are based on a requirement for 95% of the posterior distribution of the proportion of the classifications to be $< .2$ or between $.2$ and $.8$, given a prior of $Beta(\alpha_0 = .5, \beta_0 = .5)$. Cutoff values for other confidence requirements can be generated using the formulas in the Theoretical Approach section or by modifying the R code in the S1 Data.

cropland classification task. The system is flexible to the needs of different collection programs, easily accommodating different desired certainly levels and classification goals. It can also accommodate asymmetrical categories, something hinted at by the difference between Fig 3B and 3C. Overall, had the presented system been in place in the original crowdsourcing campaign which was the source of the analyzed data, this would have allowed around 2.5 times as many images to have been classified with no additional volunteer effort. That 93% of initial ratings of an image eventually agreed with that image's majority classification suggests that stronger priors would be appropriate, at least for this campaign, and would have further increased the efficiency gains described here. The potential benefits demonstrated in this case study depend on particular details of how the Cropland Capture game was implemented, including the difficulty of the tasks and the distribution of how frequently individual images were assigned. However, we have no reason to suspect that the benefits seen here are an extreme case in either direction.

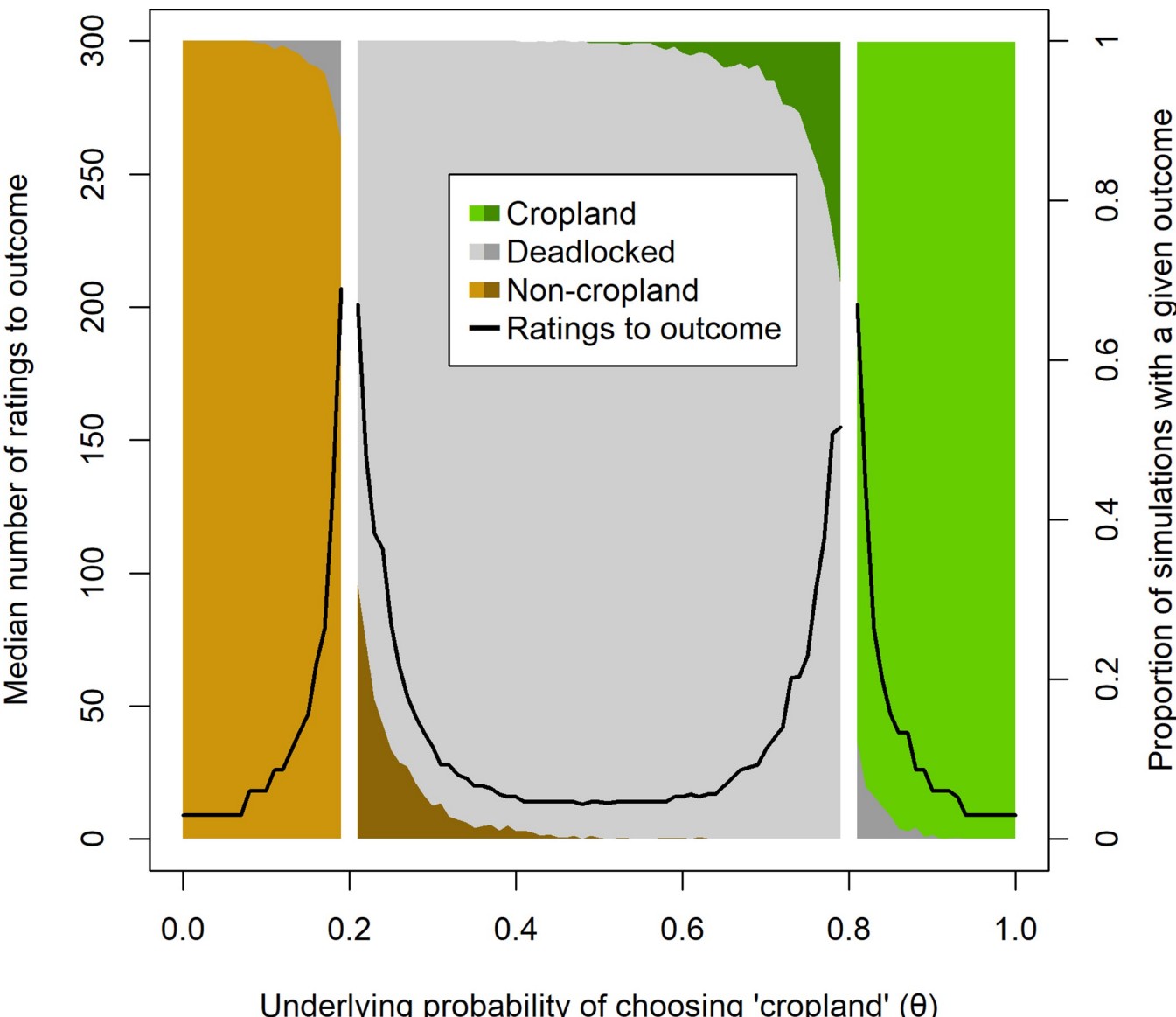

**Fig 6. Simulated classification outcomes as a function of underlying probability of choosing a cropland rating.** Colored regions represent each of the three outcome categories (cropland, deadlocked, non-cropland) and black lines the median number of ratings to reach these outcomes. The lighter colors indicate correct conclusions and the darker colors incorrect conclusions. Note that the simulations were not performed for values of $\theta$ = .2 or .8 (see main text), hence the breaks in the figure at these values.

Both confidence level parameters used in the demonstrations here can easily be modified to the needs of a particular study. Furthermore, they do not need to be symmetrical. For instance, if an intended application of the classifications is particularly harmed by a certain type of error (false positive or false negative), then one class can be given stricter requirements to be met in deciding to end collection of further classifications of a task. It is possible to go even further, for instance if a study requires identifying true positive classifications with high confidence, but is uninterested in negative classifications. In this case, the methods outlined here can be modified to identify and remove any task where some percentile of the posterior distribution exceeds the target disagreement rate. This approach would efficiently remove any image from circulation that is unlikely to reach the required standards.

A key benefit of the method outlined here is that it allows volunteer effort to be used more efficiently. Even though volunteer work is by definition free, it is still a limited resource from the perspective of users of the resulting data. Making better use of volunteer time is of clear benefit to not only the initiators of crowdsourcing and VGI campaigns, but also to the volunteers. Volunteers deserve to have their contributions used in an effective way, and may work more or better if they know that steps are being taken to use their efforts as efficiently as possible. Decreasing repetition of tasks and introducing more (and more variety of) tasks may help motivate volunteers and also help reduce volunteer fatigue, potentially yielding further gains in campaign efficiency [27]. These benefits can extend beyond mere efficiency gains. For instance, in post-disaster contexts, prioritizing images that are more likely to provide useful information to emergency responders increases the value of volunteer work and may even help save lives [28].

While the benefits of the system developed here are clear, it is also important to keep its limitations in mind. First of all, this system is intended for crowdsourced classification. In many geospatial tasks, machine learning approaches to object detection, classification, and related tasks are sufficient, eliminiating the need for such approaches. Thus, users of this method should first consider whether a hybrid or completely automated approach is more suitable [29, 30]. Second, this system is designed for a very simple task, classifying images into one of two possible categories. It has already been modified for multinomial tasks within the next generation Picture Pile application, for instance to classify images into crop types where there are several possible choices [29]. Picture Pile also supports continuous variables like estimating the degree of wealth from street-level photographs [31], which could be accomodated with different types of prior distributions. Our system also makes no consideration of rater-specific characteristics, either external factors such as where in the world they are from (and thus might be more familiar with cropland characteristics), or internal factors such as the skill they have shown at classifying images within the activity. We consciously decided not to include rater-specific characteristics because a previous study using the data analyzed here showed only tiny differences due to raters' potential local familiarity with images or their professional background in relevant disciplines [25]. Thus, it was unlikely that the added complexity of including user-specific details would have brought any benefit. However, such factors should be carefully considered before being disregarded in other tasks, particularly in the geospatial realm, and can be incorporated within a Bayesian framework [32, 33]. Further, increasingly stringent data protection laws like the European Union's General Data Protection Regulation (GDPR) complicate collection of personal information. In the case of Cropland Capture, many analyses that we performed had not been thought of at the time of the game. This makes fully informing volunteers about how their information will be used very difficult.

Although we are unaware of any previous published system to optimize the efficiency of binary task classification in VGI, some related studies provide relevant points for comparison. A recent study of mammal identification from photo trap images [14] exceeded an average confidence of 99% with only nine images. In our own study, it took a minimum (rather than mean) of nine ratings to reach a lower degree of certainty in the classification. The relative ease of achieving confidence in that study [14] may counterintuitively be due to the 21 possible mammals that volunteers had to distinguish among. This means that it takes relatively few consistent identifications of an image to conclude that the consistency is unlikely due to chance or random guessing than in our task which had only two possible choices. A remarkably similar number of volunteer contributors was needed to reach a stable positional error magnitude in a very different VGI task, the mapping of road networks in England's section of OpenStreetMap [22]. They found that although additional volunteer contributions on an

object initially improved spatial accuracy, there was on average little improvement past 10 volunteers, and none past 15 volunteers.

The system described in this study suggests several design practices for VGI campaigns. Our simulations show that for images with underlying probabilities of class membership are close to our pre-defined critical values, a decision may never be reached (see Fig 4). It thus may be useful to constrain the maximum number of times allowed for image identification before it is declared unclassifiable. Also, if possible, it is useful to obtain a rough estimate of the frequency of the two outcomes of the classification task. This may be a number that is a byproduct of the population from which the tasks are drawn, or it could be something that is intentionally targeted, for example in a campaign that seeks to positively identify instances of a certain phenomenon. This information is valuable because it could justify using more informative priors, further expediting accumulation of certainty about a task's true classification. Alternatively, these priors could be updated dynamically during the campaign as information on the relative frequency of classifications is refined. While the system presented here is based on binomial classifications, it should also be possible to generalize it to multinomial problems with more than two discrete outcomes. An alternative is to use a dichotomous approach to multinomial classification, whereby the identification task is broken into multiple binary classification tasks, similar to the dichotomous keys used by biologists to identify species. This approach might bring further advantages by reducing the cognitive load on volunteers.

## Supporting information

**S1 Data. Code for the R programming language for all analyses, calculations and figures presented in this manuscript.**
(R)

## Acknowledgments

We would like to thank the many volunteers who contributed their efforts to generating the data analyzed here.

## Author Contributions

**Conceptualization:** Linda See, Tobias Sturn, Steffen Fritz.

**Data curation:** Carl Salk, Tobias Sturn.

**Formal analysis:** Carl Salk, Elena Moltchanova.

**Funding acquisition:** Linda See, Ian McCallum, Steffen Fritz.

**Investigation:** Carl Salk, Tobias Sturn.

**Methodology:** Carl Salk, Tobias Sturn.

**Project administration:** Steffen Fritz.

**Resources:** Ian McCallum, Steffen Fritz.

**Software:** Carl Salk, Tobias Sturn.

**Supervision:** Steffen Fritz.

**Validation:** Tobias Sturn.

**Visualization:** Carl Salk, Elena Moltchanova.

**Writing – original draft:** Carl Salk, Elena Moltchanova, Linda See, Ian McCallum.

**Writing – review & editing:** Carl Salk, Elena Moltchanova, Linda See, Tobias Sturn, Ian McCallum, Steffen Fritz.

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
