## [Decision Letter · Decision Letter 0]

22 Nov 2021

PONE-D-21-13048Optimizing volunteer contributions in binary geographical classification tasksPLOS ONE

Dear Dr. Salk,

Thank you for submitting your manuscript to PLOS ONE. After careful consideration, we feel that it has merit but does not fully meet PLOS ONE’s publication criteria as it currently stands. Therefore, we invite you to submit a revised version of the manuscript that addresses the points raised during the review process.

We look forward to receiving your revised manuscript.

Kind regards,

Anwar P.P. Abdul Majeed

Academic Editor

PLOS ONE

Journal Requirements:

2. Please include the data sources used in the Data availability statement and Methods section.

"This research was supported by an International Institute for Applied Systems Analysis postdoctoral fellowship to CFS and the project ‘Using Crowdsourcing and Gaming Approaches for EO4SD Services (GAME.EO)’ funded by the European Space Agency (contract no. 4000125186/18/1-NB) to SF. The funders had no role in study design, data collection and analysis, decision to publish, or preparation of the manuscript."

We note that one or more of the authors is affiliated with the funding organization, indicating the funder may have had some role in the design, data collection, analysis or preparation of your manuscript for publication; in other words, the funder played an indirect role through the participation of the co-authors. If the funding organization did not play a role in the study design, data collection and analysis, decision to publish, or preparation of the manuscript and only provided financial support in the form of authors' salaries and/or research materials, please do the following:

a. Review your statements relating to the author contributions, and ensure you have specifically and accurately indicated the role(s) that these authors had in your study. These amendments should be made in the online form.

b. Confirm in your cover letter that you agree with the following statement, and we will change the online submission form on your behalf: 

“The funder provided support in the form of salaries for authors [insert relevant initials], but did not have any additional role in the study design, data collection and analysis, decision to publish, or preparation of the manuscript. The specific roles of these authors are articulated in the ‘author contributions’ section.

5. We note that Figure 2 in your submission contain satellite images which may be copyrighted. All PLOS content is published under the Creative Commons Attribution License (CC BY 4.0), which means that the manuscript, images, and Supporting Information files will be freely available online, and any third party is permitted to access, download, copy, distribute, and use these materials in any way, even commercially, with proper attribution. For these reasons, we cannot publish previously copyrighted maps or satellite images created using proprietary data, such as Google software (Google Maps, Street View, and Earth). For more information, see our copyright guidelines: http://journals.plos.org/plosone/s/licenses-and-copyright.

6. Please upload a copy of Supporting Information SI2 [SI2 ratings.csv] which you refer to in your text on page 21.

Reviewers' comments:

Reviewer's Responses to Questions

**Comments to the Author**

1. Is the manuscript technically sound, and do the data support the conclusions?

Reviewer #1: Yes

Reviewer #2: Yes

2. Has the statistical analysis been performed appropriately and rigorously? 

Reviewer #1: Yes

Reviewer #2: Yes

3. Have the authors made all data underlying the findings in their manuscript fully available?

Reviewer #1: Yes

Reviewer #2: No

4. Is the manuscript presented in an intelligible fashion and written in standard English?

Reviewer #1: Yes

Reviewer #2: Yes

5. Review Comments to the Author

Reviewer #1: This paper presents a method for optimizing volunteer contributions by determining whether a binary classification task has reached an acceptable level of certainty and further ratings are no longer necessary. Such a method allows the tasks meeting this threshold to be removed from the task pool, and enables volunteers to better use their time for other tasks. The authors presented the methodological details and used a image classification dataset to demonstrate the volunteer efforts that could have been saved by the their method. This paper is organized in a clear structure, and the authors did a great job in explaining their method. Some minor suggestions are provided as below:

- Introduction: The authors could add a brief discussion on a highly relevant paper:

Hu, Y., Janowicz, K., & Couclelis, H. (2017): Prioritizing disaster mapping tasks for online volunteers based on information value theory. Geographical Analysis, 49, 175–198

This reference could also help the authors make the point that a better use of volunteers' time can even save lives in urgent situations such as disaster response.

- Lines 126-128: "Consider a hypothetical land cover classification task where four volunteers say they see cropland and a fifth does not detect cropland. In this case, 20% of the ratings disagree with the majority." The authors might want to revise this sentence to "... 20% of the *total* ratings disagree with the majority." for clarity; otherwise one may wonder whether the rate should be 25% (i.e., 1 no cropland and 4 cropland).

- One important assumption of the method presented is that the ratings are independent, i.e., one volunteer's judgement should not be affected by those of other volunteers. Such an assumption may not necessarily hold in voluntary tasks. For example, some tasks may allow one to see previous ratings; or there can be an online forum where volunteers can discuss the tasks. The authors may need to explicitly state this assumption so that users of your method are made aware of it.

- The experiments are based on one particular dataset. Some more information about the volunteers (e.g., how these volunteers were recruited) should be provided. The authors did provide some relevant references, but having some more information about the volunteers can make this current paper self-contained.

Reviewer #2: Dear Authors,

thanks for submitting this article. Overall it is well written and covers an interesting topic. I have a few comments for things you might be able to improve.

First, starting with the data availability. Maybe it's just that I didn't get the right files forwarded from the editorial manager or overlooked it. Is the data you used for this research somewhere available? Ideally you could put in on some open sharing platform, as the data might be an interesting field of study also for other scholars.

The second major concern at the current state of the manuscript is the discussion. From my point of view you should try to improve this section, for instance by better comparing your results to the results of other studies. In your current manuscript you only cite a few references. Some references you use in the discussion (e.g. [22] about OpenStreetMap) are not directly comparable to the kind of crowdsourcing you are investigating in your study. Furthermore, it would be good if you can provide more insights how your proposed method compares to other ways of improving the crowdsourcing approach, e.g. remote sensing based approaches, object detection etc. also offer some potential to reduce the overall number of volunteered classifications needed. Whereas it is clear that this is not part of your analysis, you could add this to your discussion.

Regarding your results I have one main difficulty in properly understanding them. Do 59.1% refer to the overall work or does it only apply to the images for which work has been removed? It might be good to add a figure which shows the overall gain of your approach, e.g. by showing the overall number of classifications and the number of classification you identified would have been needed. The 59% can be misleading somehow as there is a different number of raters per image and this number doesn't follow a clear pattern. (Please correct me if I'm wrong, but this is what I remember from one of your previous papers)

In the discussion you also state that your approach would "have allowed around 2.5 times as many images to have been classified with no additional volunteer effort." (L332). Maybe it's obvious but could you highlight again how this 2.5 are derived. You mention it in the results section as well. Maybe there you can elaborate more.

Finally, it might be good to further highlight some limitations of your approach. In your discussion section you don't mention any, but I'm pretty sure that there will be some. In you proposed approach (but similar also for simple majority agreement) you don't consider user characteristics or the location of the tasks. Both aspects might also have an impact on the classification quality. Could these additional information be included in an approach such as you presented?

Some quick comments regarding the Figures: It might be good to add a legend to Figure 1 and Figure 3 so that readers can understand easier what the colors refer to. In Figure 4 there seems to be a pattern (like a seasonality) in the data. how can this be explained? Is this a limitation of the proposed method?

Another quick one about "disagreement rate" (L125). Is this your own definition or a commonly agreed one? I think it makes sense, but are not sure where the definition comes from.

Finally, a more personal question. You show interesting results. Still the number if ratings needed for such a "simple" task seems rather high to me. For instance, when mapping settlements with MapSwipe one usually uses much less number or ratings per image. Usually between 3-5. Is this a "problem" or is the task of identifying buildings just a bit easier than identifying cropland?

Thanks for your work,

Best regards

6. PLOS authors have the option to publish the peer review history of their article (what does this mean?). If published, this will include your full peer review and any attached files.

Reviewer #1: No

Reviewer #2: **Yes: **Benjamin Herfort

---

## [Author Response · Author response to Decision Letter 0]

7 Feb 2022

Please see point by point responses to the reviewers comments in the uploaded file.

---

## [Decision Letter · Decision Letter 1]

4 Apr 2022

How many people need to classify the same image? A method for optimizing volunteer contributions in binary geographical classifications

PONE-D-21-13048R1

Dear Dr. Salk,

We’re pleased to inform you that your manuscript has been judged scientifically suitable for publication and will be formally accepted for publication once it meets all outstanding technical requirements.

Kind regards,

Anwar P.P. Abdul Majeed

Academic Editor

PLOS ONE

Additional Editor Comments (optional):

Reviewers' comments:

Reviewer's Responses to Questions

**Comments to the Author**

1. If the authors have adequately addressed your comments raised in a previous round of review and you feel that this manuscript is now acceptable for publication, you may indicate that here to bypass the “Comments to the Author” section, enter your conflict of interest statement in the “Confidential to Editor” section, and submit your "Accept" recommendation.

Reviewer #1: All comments have been addressed

Reviewer #2: All comments have been addressed

2. Is the manuscript technically sound, and do the data support the conclusions?

Reviewer #1: Yes

Reviewer #2: Yes

3. Has the statistical analysis been performed appropriately and rigorously? 

Reviewer #1: Yes

Reviewer #2: Yes

4. Have the authors made all data underlying the findings in their manuscript fully available?

Reviewer #1: Yes

Reviewer #2: Yes

5. Is the manuscript presented in an intelligible fashion and written in standard English?

Reviewer #1: Yes

Reviewer #2: Yes

6. Review Comments to the Author

Reviewer #1: (No Response)

Reviewer #2: I am satisfied that my suggestions have been addressed sufficiently, and I recommend that the article is published.

7. PLOS authors have the option to publish the peer review history of their article (what does this mean?). If published, this will include your full peer review and any attached files.

Reviewer #1: No

Reviewer #2: **Yes: **Benjamin Herfort

---

## [Editor Report · Acceptance letter]

3 May 2022

PONE-D-21-13048R1 

How many people need to classify the same image? A method for optimizing volunteer contributions in binary geographical classifications 

Dear Dr. Salk:

I'm pleased to inform you that your manuscript has been deemed suitable for publication in PLOS ONE. Congratulations! Your manuscript is now with our production department. 

Kind regards, 

on behalf of

Dr. Anwar P.P. Abdul Majeed 

Academic Editor

PLOS ONE